# Becoming Safe, Legal, Mature, Moderate, and Self-Reflexive: Trajectories of Drinking and Abstinence among Young People

**DOI:** 10.3390/ijerph19063591

**Published:** 2022-03-17

**Authors:** Eva Samuelsson, Jukka Törrönen, Josefin Månsson, Filip Roumeliotis

**Affiliations:** 1Department of Public Health Sciences, Stockholm University, 106 91 Stockholm, Sweden; jukka.torronen@su.se; 2Department of Social Work, Stockholm University, 106 91 Stockholm, Sweden; josefin.mansson@socarb.su.se; 3Department of Criminology, Stockholm University, 106 91 Stockholm, Sweden; filip.roumeliotis@criminology.su.se

**Keywords:** youth, alcohol, abstinence, qualitative longitudinal data, actant, assemblage, trajectory, chains of translation

## Abstract

In recent years, a vast body of research has investigated trends of declining alcohol consumption among youths. However, the extent to which restrictive-youth approaches towards drinking are maintained into adulthood is unclear. The aim of this study is to explore how young people’s relation to alcohol changes over time. Our data are based on longitudinal qualitative in-depth interviews with 28 participants aged 15 to 23 conducted over the course of three years (2017–2019). The study draws on assemblage thinking by analysing to what kinds of heterogeneous elements young people’s drinking and abstinence are related and what kinds of transformations they undergo when they get older. Five trajectories were identified as influential. Alcohol was transformed from unsafe to safe assemblages, from illegal to legal drinking assemblages, from performance-orientated to enjoyment-orientated assemblages, and from immature to mature assemblages. These trajectories moved alcohol consumption towards moderate drinking. Moreover, abstinence was transformed from authoritarian assemblages into self-reflexive assemblages. Self-control, responsibility, and performance orientation were important mediators in all five trajectories. As the sober generation grows older, they will likely start to drink at more moderate levels than previous generations.

## 1. Introduction

In recent years, a vast body of research has searched for explanations that indicate the trends of declining alcohol consumption among youths. No single explanation is sufficient to understand why such a large share of youths in high-income countries today choose to abstain from alcohol compared to previous generations [1,2]. Earlier, there has been a close link between drinking to intoxication among 15-year-olds and the general patterns of drinking to intoxication in the older population [3]. When youth drinking has decreased so strikingly, it is relevant to ask whether emerging adults’ alcohol habits will be relevant as the “sober generation” grows older [4]. In this study, we follow a group of Swedish youths over the course of three years to qualitatively explore how their relationship to alcohol changes over time.

The clear decrease in youth alcohol consumption during the past two decades is a cross-national phenomenon in high-income countries, which confirms the globalisation of youth culture [5]. To understand these trends, broader explanations across and within countries are needed [6]. In Europe’s Nordic countries, youths have previously displayed comparably high levels of heavy drinking. Drinking to intoxication has been described as a central rite of passage into adulthood [7], a behaviour that the large majority “mature out of” as they grow older [8] and need to meet the standards of responsibility in studies, work, and family. Nowadays, the onset age of drinking is substantially delayed, and more youths tend to decline alcohol altogether. Alcohol seems to have lost its previous social status [9] and is not seen as an important tool in youth identity exploration as seen earlier [10]. Shifts in parental practices have been suggested as one explanation for reduced drinking among youths [2,11]. Today’s adolescents are less inclined than prior generations to have sex, date, drink alcohol, work, go out without parents, and drive a car, which have all been interpreted as signs of delayed adolescence. Youths tend to live according to slower life strategies in contexts with more engaged parenthood, prolonged education, and delayed reproduction [12].

Other research, however, proposes that decreased risk-taking such as heavy drinking is rather a sign of increased diversity when engaging in adulthood [5] or in advanced adulthood, an early maturation of young people [13]. Today’s youth are expected to be rational, responsible, and actively invested in their studies and work opportunities, building social networks in clubs, sports, and social media to succeed in life [4,10]. To consume alcohol in a moderate or excessive way can be seen as a means to achieve status in some social circles, but considering the high demands of performance in everyday life, drinking to intoxication and losing control is not generally seen as rational behaviour. There are also signs of increased mental health problems among Swedish youths, possibly related to structural changes in the school and labour markets resulting in increased competition and segregation [14,15].

Alcohol consumption among 15- to 16-year-olds is well-studied, but knowledge on the extent to which modest drinking practices among youths will sustain later in their lives is scarce [16]. The proportion of the population aged 16 to 29 years old with risk consumption of alcohol (drinking to intoxication and/or 14 units per week for men and 9 units per week for women) has declined from 37 percent in 2004 to 19 percent in 2021 [17]. However, the circumstances that could explain these changes remain unclear. Despite similar developments in young people’s alcohol consumption in high-income countries, the decline in risky drinking among 18- to 24-year-olds has continued in Australia, whereas in Sweden the decrease primarily took place before 2010 [18]. To understand how young people’s alcohol habits are related to overall societal changes and changes in young people’s everyday life practices, concerns, and lifestyles [19], more qualitative research is needed [1]. Longitudinal qualitative research has the potential to be a powerful approach to understand the complexities of youth alcohol consumption across time [20]: including relationships with peers and family, the experience of intoxication, the influence of leisure time activities, health, school, and work opportunities. The study thus explores how young people’s relationship to alcohol changes over time by looking at how it is affected by multiple possible interacting factors.

### Theoretical Approach

Young people’s drinking or abstinence is influenced and mediated by multiple interacting causes [21]. When we follow the development of young people’s drinking or abstinence over time, we, therefore, need to pay attention to what kinds of heterogeneous elements it is associated with, how these elements enable, hinder, increase or decrease young people’s capacities to drink or avoid drinking, and what kinds of transformations they undergo when young people get older. The concepts of “assemblage”, “actant”, “trajectory”, and “chains of translations” are important in this kind of research task.

An *assemblage* refers to heterogeneous sets of relations composed of both human and non-human actors and agencies [22,23]. Drinking or abstinence assemblages, for example, are constituted from a mix of physical, emotional, and social relations that impact variously how much and in what way drinkers consume alcohol or remain sober [24]. Who or what acts in a drinking or abstinence assemblage is called an *actant* [22]. Human and non-human actors become actants when they modify or alter the course of action. Actants’ agency is relational. They can only move action forward as part of an assemblage, as linked to other actors, through the relations in which they can enable, hinder, increase, or decrease actors’ capacities to drink or stay abstinent. A *trajectory* describes how a certain activity such as drinking or abstinence evolves over time into a particular temporal narrative in which its main actants undergo *chains of translations* as they pass through diverse events, encounters, and relations [22,25]. As our longitudinal interview material captures the maturing process of emerging adults, it provides expressive data to follow what kinds of trajectories the development of their drinking and abstinence form and what kinds of chains of translation their main actants and assemblages undergo as our participants age. In this kind of analysis, the development of alcohol consumption or abstinence is approached not as an outcome of individual motives, intentions, or decisions but as an outcome of heterogeneous sets of relational agencies and practices that condition and guide its movement toward specific directions [26].

## 2. Materials and Methods

### 2.1. Recruitment Procedure

This study was conducted within a broader longitudinal project examining declining youth alcohol consumption in Sweden [13]. In all, 28 young people were interviewed over the course of three years (2017, 2018, and 2019). Participants were recruited from various secondary and upper secondary schools in Stockholm County and mid-Sweden areas with assorted sociodemographic compositions, as well as through non-governmental organisations and social media ads. In the recruitment process, efforts were made to reach a heterogeneous sample in terms of sociodemographic factors and drinking practices (see [10] for more information on this purposive sample, and Appendix A for descriptive details of the participants).

### 2.2. Participants

The data consist of 28 complete cases of three interviews conducted when the participants were aged either 15–17 to 17–19 or 18–19 to 20–21 years. The majority of those interviewed, 19 out of 28, were girls. Nine were born abroad or had parents born abroad. When they were recruited, twelve of the participants were secondary school students and had thus started upper secondary school in wave 3. Sixteen participants were in the last year of upper secondary school in wave 1, which meant that at the time of the third interview, they had started university studies, were looking for employment, or had begun working full-time or part-time. During the study, the participants had also been subject to other notable life transformations—turning 18 (the legal age of obtaining a driver’s licence and buying alcohol in bars and restaurants in Sweden) and turning 20 (the legal age of buying alcohol in retail stores in Sweden), meeting partners, making new friends, and moving away from home. In the third and last interview, eight participants were still in upper secondary school, thirteen had started university studies, five were working full-or part-time, and two were unemployed. Based on self-reports, the participants were categorised into abstainers, moderate drinkers (avoiding intoxication) or heavy drinkers (drinking to intoxication, more than six units per occasion, at least once a month). As displayed in Table 1, the proportion of abstainers decreased from wave 1 to wave 3, and the proportion of heavy drinkers increased from wave 2 to wave 3. This change is due to the fact that participants from the younger cohort to a greater extent were abstainers in wave 1. In comparison with population level data, the proportion of abstainers in wave 3 is quite similar, but the proportion of moderate drinkers is somewhat lower and the proportion of heavy drinkers is somewhat higher [17].

### 2.3. Interviews

The interviews were conducted primarily face-to-face in wave 1, via Skype in wave 2, and over the phone in wave 3. After obtaining informed consent, the interviews were audio-recorded and lasted 30 to 99 min (average 55 min). In the first wave, five interviews were held with pairs of friends, as per their wishes. In the second wave, four interviews were held in pairs. In the third wave, all interviews were held individually, resulting in a total of 75 interviews. The semi-structured interview guide included open questions covering themes such as: leisure-time activities; friends; social media use; drinking behaviours and occasions; narcotics and tobacco use and perceptions around; gambling and gaming; understandings of health; having fun drinking or abstinence; and parents’ and siblings’ supervision. Prior to each interview occasion, in waves 2 and 3, we processed previous interview(s) to be able to pose relevant follow-up questions regarding different life aspects of the individual youths. The interviews were conducted by the first, third, and fourth authors. When possible, we tried to ensure that the same author interviewed the same youth several times to create a safe environment and maintain an alliance. As compensation, the participants were offered cinema tickets. The interview procedures followed the current ethical guidelines emphasising voluntariness, confidentiality, and informed consent.

### 2.4. Coding

The recorded interviews were transcribed verbatim by a transcription company. After each interview, we wrote brief analytical memos with a specific focus on changes in drinking and abstinence practices and habits. Then, the interviews were coded by using the NVivo software program. In the coding, we first paid attention to how the interviewees described their relation to and reasons for drinking and abstinence, the practices to which they related their drinking and abstinence, and how these practices were affected by other everyday life practices. Moreover, we followed how our participants’ relations to these issues changed over time. Through this coding process, we identified five types of trajectories that play an important role in young people’s maturation process in relation to abstinence and drinking.

### 2.5. Analysis

In the subsequent analysis, we drew on the concepts of “assemblage”, “actant”, “trajectory”, and “chains of translations” as explained above and these are displayed in Supplementary Material S3 for a comprehensive overview. This analysis disclosed that the main actant in the first trajectory is “drinking place” (e.g., codes such as private vs. public venues). This trajectory describes how teenagers’ private drinking places are first related to unsafe relations and assemblages that undergo diverse chains of translations towards safe relations when their drinking moves to public drinking venues. The main actant in the second trajectory is “age” (e.g., codes such as turning 18, becoming mature). This trajectory deals with how young people’s drinking and abstinence undergo diverse chains of translations in relation to legal age. The main actant in the third trajectory is “performance” (e.g., codes such as responsibility, conscientiousness). In this trajectory, performance-oriented assemblages may undergo chains of translations that make room for enjoyment-emphasising moderate drinking as participants get older. The main actant in the fourth trajectory is “intoxication” (e.g., codes such as negative consequences from drinking, health concerns). This describes what kinds of chains of translations intoxication meets by the ageing of our heavy drinkers. The fifth trajectory deals with “abstinence” (e.g., codes stating reasons to abstain such as religion or political values). It describes the movement of abstinence from authoritarian-related assemblages to more self-relying assemblages. All these maturing trajectories are related to becoming a controlled, responsible, and adult-like actor [4,10]. In what follows, we present in more detail these trajectories and their actants, assemblages, and chains of translations.

## 3. Results

### 3.1. Drinking Becomes Translated from Unsafe to Safe Assemblages

The first maturing trajectory describes how drinking becomes unlinked from an assemblage of unsafe drinking places and relations and transformed into an assemblage of safe drinking. The unsafe drinking assemblage is built around underage drinking places, events, and encounters. In it, drinking bodies become related to crowded and loud home parties, excessive alcohol consumption, chaotic circumstances, emotional instability, lack of control, and vulnerability. This all changes when drinkers gain access to public drinking venues when they turn 18. At this point, their drinking places, events, and encounters undergo chains of translations through which their drinking becomes a tamed, less dangerous activity related to moderate consumption, emotional stability, control, and absence of risk. In addition to drinking places, an important actant in this trajectory is the legal age, which guarantees access to public drinking venues. Tara’s quotation below exemplifies the tamed and safe character of public drinking places:


*I think it is because you are getting older and are allowed to go out and drink at the club. It’s not that scary anymore. Since everybody else is doing it, it shouldn’t be dangerous (Tara, 18 years old, moderate drinker).*


Similarly, Santiago describes how unsafe drinking places and relations transformed into safe ones after gaining access to public drinking venues. In drinking assemblages, the entrance into adulthood introduces new actants such as legal drinking venues (instead of home parties) stabilised by alcohol policy and legal repercussions, public control (instead of peer control), state-regulated beer and wine (instead of hard liquor bought from dealers), and responsibly behaving consumers (instead of youngsters who lack the capacity to control their drinking).


*Before 18, one couldn’t drink at the bars and instead bought from a dealer. And that, I would say, is irresponsible because you never know what it contains. I know a lot of friends who thought it wasn’t cool, but they found it fun to drink, and they did it a lot. But after they turned 18, they changed. They went to the bars instead and didn’t drink as much. (…) It feels like they drink more responsibly and less now. (…) People stopped going to home parties (Santiago, 19 years old, moderate drinker).*


In this safe drinking assemblage, the position of alcohol as an actant transforms from playing a central role in the unsafe drinking assemblage into possessing a more peripheral agency. For example, when Sophie gets older, and her drinking becomes translated as part of a more secure and stabilised assemblage at university, her drinking takes the backseat, and other actants become the driving forces of the situation, e.g., celebrating the hard work achieved during the week and the moment itself.


*I would not say that it [the alcohol] is that important as such. It’s more like you have gotten accustomed to that it’s always around you. For example, I tend to go to all the Friday pubs, not that, “Oh, it’s Friday, let’s get drunk” but more like, “Ah, it’s Friday, a week of studies is over”, and I’ll hang out with all the students (…) and just celebrate the weekend. And then I have two or three beers because I’m there; it’s there. And that’s nice (Sophie, 21 years old, heavy drinker).*


### 3.2. Drinking Becomes Translated from Illegal to Legal Assemblages

As seen in the section above, turning 18 years old is one central actant in translating unsafe drinking into safe drinking. It is, however, also pivotal in the trajectory where legal aspects of drinking are highlighted in comparison to other elements in drinking. For example, Ella does not even consider drinking as a possibility before she turns 18. She argues that this is not a question of choice but a matter of whether someone has a right to drink or not. Since she and her friends are minors, they simply do not drink.


*Not that we are “abstainers”, but more like “we don’t drink”. We are minors—why would we? It’s as if there is a totally different perception of what’s allowed compared to earlier. (…) It feels like many people take them [the age limits] more seriously (Ella, 15 years old, abstainer).*


The legal age of 18 defines the moment a person gains the right to drink; it distinguishes a minor from an adult. As this moment is upheld by legal institutions, it moves the actors who turn 18 towards an adult identity with corresponding rights, such as permission to develop a relation to drinking. Our three interviews with William exemplify the power of this legal age limit. At age 15, William finds drinking “stupid” and prefers to spend his time playing video games and sports. At age 17, he is still an abstainer but plans to start drinking when he turns 18. His action is bolstered by strict parental rules.


*They [the parents] say I shouldn’t start drinking… [and that I should wait until] I turn 18 (William, 15 years old, abstainer).*



*When I’m 18, I can drink, and they will trust me. (…) I want to devote myself to sports, and I can’t do that if I drink a lot. When I’m 18, or at graduation, there will be a champagne breakfast and such. Then I’ll probably drink. (…) First of all, I’ll have turned 18 and [at that point] we can celebrate something (William, 17 years old, abstainer).*


William’s decision to consider drinking only after turning 18 is thus an outcome produced by multiple actants such as legal age, maturation, and trust of parents’ advice and rules.

Many of our participants who turn 18 consider the translation of abstinence into drinking—especially into moderate consumption of alcohol—as a natural step in becoming an adult. At 18, you can enter into assemblages where alcohol functions as an actant that increases your capacities in diverse social events. You can go to bars and clubs (Victoria), partake in campus drinking activities (Lily, Caroline, Sam), seek after-work drinking occasions (Emily), and partake in adulthood by joining your parents for a glass of wine (Alice). Olivia’s maturing process exemplifies this. When she turned the legal age of 18 and was allowed to buy and consume alcohol, her abstinence was translated into moderate drinking, which she considered a natural process. The translation increased her capacities to engage in adult-like sociability, “like having coffee with a friend at night, but drinking wine instead.”

### 3.3. High-Performing Assemblages Exclude Drinking or Become Translated to Include Drinking for Pleasure

The third type of maturing trajectory concerning drinking deals with the opposition between assemblages of high performance and self-control on the one hand and assemblages of hedonism or pleasure on the other. In the high-performance assemblage, the actors aim to achieve excellent grades in school, aspire to successful careers, or perform at the top level in sports. Their focus on performance is boosted by both their parents’ and their own expectations to succeed. Sana’s quotation below exemplifies this assemblage and shows how it is often linked to responsibility towards one’s own parents.


*My parents would prefer me to do something they never had the opportunity to do, and that is to study and educate myself. (…) I try to prove myself to my parents; I want them to perceive me as a very responsible person. I want to make my own money, and I want them to trust me (…) because you never know what will happen. One of them could pass away, or we could lose the house and then it’s always good if I have money on the side to provide for us for at least a month or so (Sana, 16 years old, abstainer).*


As Sana grows older, her performance orientation becomes more stressful as her friends and boyfriend expect her to spend more time with them and not so much time on her studies. Sana solves these conflicting expectations by continuing her sobriety and realising that in the future, she can participate in drinking situations sober while pretending to drink.

Alice’s maturing trajectory, in turn, demonstrates a case in which the interaction between different assemblages does not lead to conflict between various expectations. Instead, the diverse assemblages fold into one another to enable new possibilities for her and enhance her capacities for pleasure and enjoyment in life. First, her engagement in elite sports—a high-performance assemblage that builds on relations of regular exercise, rules of abstinence, self-control, lack of time, and uncomfortable feelings in drinking settings—hinders drinking and makes it appear unattractive and unhealthy.


*I have never had the feeling of wanting to try it out [drinking]. I think it seems scary to lose control, sort of, and I don’t like those settings. (…) It seems uneasy. (…) I can’t see a reason for it. But, well, it has to do with health, too, why I wouldn’t (Alice, 18 years old, abstainer).*


One year later, her high-performance assemblage (where she now combines elite sports with a job) still keeps her abstinent, but she recognises that drinking might be fun.


*First, I thought it could be fun, but then I had to work early the morning after, and I had practice. I was considering it, but it didn’t happen. Well, I don’t know, I might be a bit more attracted to it (Alice, 19 years old, abstainer).*


In the last interview, however, she describes how her high-performing assemblage has undergone chains of translations that have made room for a new relation to drinking. With the influence of her friends and sister, she has started to drink moderately and learnt in the process that alcohol as an actant does not necessarily diminish her body’s capacities for action and well-being but rather—when consumed in the right and controlled way—increases them so that she can better enjoy the encounters with people, food, drinking settings, dancing, and herself.


*I have a friend, and when I got to know him more, we started to hang out and drink beer together. (…) I have realised that it is a way to meet people in society today (…) I think one beer tastes good, it is a nice beverage with food, sort of. (…) It feels extra luxurious (...) I become a bit more relaxed, and things can be a bit more fun. (…) On occasion, I have felt this feeling of panic in alcohol environments, that I am not comfortable in the situation (…) I just want to leave. (…) I have managed to overcome that a little. My sister knows how I feel, so she tried saying to me, “Come on”, tried to make me dance and eventually she managed to and then I overcame [my fear] … I went from feeling really uncomfortable to actually enjoying myself (Alice, 20 years old, moderate drinker).*


### 3.4. Heavy Drinking Becomes Translated from Immature to Mature Assemblages

The fourth type of trajectory in relation to drinking deals with the opposition between immature and mature intoxication. The assemblage of heavy drinking becomes—as time passes—modified by linkages that increase heavy drinkers’ capacities to drink in a more controlled, responsible, and personal way. Oliver describes this in the following way:


*It feels more fresh now, sort of. Before [the drinking] could be more frowzy, but now… I have more control over myself. That has changed. But I do the same things. (…) Take more responsibility, am older and hang out with older people. I don’t have to go out partying just because everybody else does it. Now I have other friends to watch a movie with. (…) I have more options of what to do. More money and more diversity. Before, I could go out drinking because I had just turned 18 and everybody wanted to go out. Now I’m 20, and I can sit and knit a sweater if I want to [laughter], and that wouldn’t be a problem (Oliver, 20 years old, heavy drinker).*


As our heavy drinking participants become older, their assemblages may undergo multiple “chains of translations” that can increase, decrease, or momentarily stabilise their drinking. For example, when Oliver focuses on his studies, this results in momentary decreases in his heavy drinking. When his romantic relationship breaks down, and he becomes single, this culminates in an increase in his drinking, and when he notices that drinking makes him look obese, this, in turn, for a while, decreases his drinking. The next excerpt from Thomas exemplifies how in the maturing process, heavy drinking habits tend to diversify through encounters with alternative drinking styles:


*I drink a lot less nowadays. (…) I know I have more control now. It’s not that I drank myself under the table every weekend, but it doesn’t happen as often that I get that drunk. And that’s because, I think, I don’t find it as fun anymore. (…) I’ve started to like red wine and beer too. (…) In that sense, I’ve started to drink in other social settings. Not just to get hammered. I drink because I enjoy it. Is it good for me?—That I don’t know (Thomas, 20 years old, heavy drinker).*


For many of the heavy drinkers, encounters with intoxication have increased their capacities to reflect on their drinking. Thus, previous negative drinking experiences have become actants moving them towards a more moderate drinking assemblage. For example, at the ages of 18 and 19, Lily first drank heavily with her girlfriends at home parties and bars—especially when she had finished upper secondary school and started working while still residing at her parents’ house; she had money to party. She describes how things often got out of hand, how her parents expressed concerns over her drinking and how she found out that her grandfather had previously experienced alcohol problems. These encounters made her reflect on her drinking. Later, she also realised that drinking caused anxiety and disturbed her studies.


*I have had anxiety [after drinking]. It’s felt unnecessary to drink because I know it will make me feel bad. I’ve had depression and take antidepressants too. Sometimes the anxiety gets worse the day after and the few days after drinking. (…) I don’t want that anymore because I don’t want to feel that way. (…) I’d rather concentrate on my studies (Lily, 20 years old, previous heavy drinker, currently moderate drinker).*


These negative experiences and her concerns over her health translated her heavy drinking trajectory into a moderate drinking trajectory. In our data, heavy drinkers describe similar kinds of translations. As they accumulate intoxication experiences, they not only translate their relationship with drinking into a more mature, responsible, and controlled one but also become more ambivalent and cautious. The learning-by-doing introduces actants to their heavy drinking assemblage, which decreases their desire to drink. They relate concerns linked to health (damages to the brain, Santiago; harm to the liver, heart, and blood vessels, Sophie; stomach aches and gastritis, Julia and Fadi), risks of drinking (sexual abuse, Tara; addiction, Emily), and appearance (losing control, gaining weight, becoming bloated, Oliver).

On the other hand, in our data, there are also participants who—despite developing an awareness concerning the risks of heavy drinking—state that for a few more years, they can continue to drink heavily, unhealthily, and irresponsibly since they are still young. For example, Thomas insists that his heavy drinking eventually will decrease.


*I don’t think I drink an awful lot, considering that I am 21 years old. I would not regard myself as an alcoholic. (…) You know it’s not good for you to drink, so you will scale down eventually. But I want to have fun now while being young, so I do it anyway (Thomas, 21 years old, heavy drinker).*


### 3.5. Abstention Is Translated from Authoritative Assemblages to Self-Reflexive Assemblages

The last trajectory concerning drinking deals with the tensions between authoritative and self-reflexive forces in abstinence. The participants’ abstinence is first conveyed by an authoritative assemblage with actants such as religious beliefs, parental rules, subcultural norms, temperance movements, or political values, but later mediated in conjunction with a more self-reflexive assemblage related to personal reasons, acknowledged feelings, and authentic individual experiences. Through these chains of translations, abstinence becomes modified into more nuanced relational forces. For example, when Adina is 18 years old, one of the driving actants in her abstinence assemblage is religion—she does not drink because she is a Muslim. Two years later, her interview illustrates that the relational agency of her religious beliefs has weakened, and her abstinence is further moved by actants that are associated with personal concerns, experiences, and choices.


*I still don’t drink (…); my thoughts about it are still the same. (…) And I have maybe developed a broader picture of why. That it’s not the best and you can have fun without drinking. Before, when I was younger, I thought mainly about religion, but when you get older and start thinking… that you are not aware of yourself and such [while drinking], that causes concerns. You get more concerned about yourself somehow. Especially if there are things happening that are not good, that you can’t trust everyone. But I discovered that later, for me, I should actually not drink. (…) I’ve reached this age and made my own decision, and I will continue with it, to not drink (Adina, 20 years old, abstainer).*


Similarly, in Mateo’s abstinence trajectory, the primary actant is first a figure of authority, his father, who argues abstinence to Mateo by maintaining that drinking would ruin his health, diet, and training schedule in sports.


*I know what my father would think about parties and such. He doesn’t approve. He would be very disappointed if I wanted to do it, but I respect that he doesn’t want me to. So I won’t. (…) My friends in school, they know me, and they know that my father doesn’t like it. So, they wouldn’t ask. (…) He tells me, “You should not drink” since he knows about my sport and that. He told me, “It would ruin your exercise and condition”. (…) If I were an adult, he wouldn’t be angry (Mateo, 16 years old, abstainer).*


One year later, Mateo interestingly finds himself in his father’s shoes and presents himself, his studies, his exercise, his risk of addiction, and saving money as actants that increase his capacities to stay sober.


*I don’t drink because my exercise would crash. It ruins your health, (…) diet and the training program that I have set up myself. I don’t find it important to drink and such. You know you get addicted and such. (…) I don’t want to take the risk. (…) It’s sort of a matter of principle that I learned myself. (…) I can focus on studying and exercising instead of partying and drinking and spending money on such things (Mateo, 17 years old, abstainer).*


## 4. Discussion

In the article, we have analysed 28 complete cases of in-depth interviews conducted over the course of three years (2017 to 2019). The study deepens our understanding of how emerging adults approach alcohol over time. As the study results are based on a small purposive sample, one limitation could be the lack of generalisability of the results to Swedish youths in general. Social desirability bias in self-reported data should also be considered. However, by including interviewees from different sociodemographic backgrounds and by interviewing the same interviewees three times, the study is able to both capture the variation in young people’s drinking and abstinence trajectories and show how these trajectories are influenced by diverse kinds of material settings, relationships, health concerns, interests, emotional issues, practices, and habits.

Our analysis identifies five trajectories that describe what kinds of chains of translations young people’s relation to alcohol have undergone over time. The first trajectory shows that young people consider underage drinking unsafe. It is related to informal spaces [27], vulnerable relations, chaos, uncertain substances, and lack of control. When our participants turn 18 years of age and gain access to public drinking spaces, drinking becomes translated into safe, reliable, and controlled forms where alcohol takes a lesser role as an actant.

The second trajectory demonstrates how the accepted time to start drinking has moved from early adolescence to the moment of becoming a legal adult. If, for the earlier generations, the subcultural norms, group pressure, and intoxication provided the “social clock” [28] for the passage into adulthood [7], for the current abstaining minors, the time of the clock is set by the law and legal age [29]. This displacement of the “social clock” for alcohol debut from the subcultural determination into official institutions indicates that young people’s subcultural heavy drinking assemblages have lost their hegemonic positions among young people. They do not have the same power as they previously did to inaugurate young people into adulthood through drinking, which is reflected in the general delayed debut of drinking among Swedish youths [30]. The first trajectory and its chains of translations also testify to this. They point out how young people today trust the influence of the regulated official drinking venues in managing their drinking practices and are critical towards the influence of the elements that emerge in their own subcultural contexts.

The third trajectory demonstrates the opposite displacement in young adults’ passages to adulthood. While the first and second trajectories exemplify in what way the current teenagers’ childhood lasts longer than that of the earlier generations [12], the third trajectory demonstrates how young people very early on are expected to become adult-like responsible actors, as entrepreneurs of themselves [31], who need to continuously reflect upon what kinds of connections they need to cherish to become successful with promising futures [10]. This third trajectory indicates how elements related to performance increasingly become actants that modify young people’s abstinence and drinking course. For some of our participants, the expectations to perform well cause stress, strengthening their dedication to remain abstinent. However, there are also participants who manage to navigate their stressful environment in a seemingly more balanced way. For these participants, moderate drinking develops into an actant that increases their capacities for enjoyment and well-being in life. It helps them in taking breaks from competition, pressure, and expectations.

The fourth trajectory also displays how in heavy drinking assemblages, the maturation process means embracing relations that increase heavy drinkers’ capacities to act responsibly, controlled, and reflexive. This trajectory further concretises how young people, through multiple encounters with intoxication, learn to realise that heavy drinking is a risky and unhealthy practice. This learning-by-doing intoxication introduces to their heavy drinking assemblage actants that link their drinking to diverse physical, emotional, and social concerns and modify their relation to drinking to ambivalent and cautious [32,33]. These chains of translations and relational agencies decrease rather than increase their drinking.

The last trajectory expresses how abstinence in emerging adulthood is transformed from authority-related ideological, religious, and parental justifications towards personally experienced reasons, and in these, chains of translations become more mediated by individualised concerns. Although research suggests that abstinence might not be as socially sanctioned among youths in some contexts [34], youth non-drinking has lately become normalized in many high-income countries [35]. This suggests that diverse reasons for abstinence have become accepted and being sober apprehended as less deviant during recent years [36].

Overall, our results point out how young people’s relation to school, family, leisure, and the labour market is influenced early on by expectations in which the individual responsibility to exercise self-control, gain self-knowledge, and implement self-improvement is emphasised as an important element in becoming an authentic and successful person [37,38]. These modifications in young people’s drinking connections and environments explain why they increasingly postpone their drinking to the legal age, and when they decide to enter drinking settings, they prefer to drink moderately. Furthermore, our analysis suggests that when young people get older and accumulate drinking experiences, they often start to see excessive undisciplined, irresponsible, or unhealthy drinking as a sign of a moral failure of the self [39]. This further orientates their drinking towards moderate or more disciplined drinking relations and practices [40]. As there are concerns of increasing mental health issues among young people [41], more research is needed to understand how diverse aspects of emerging adults’ everyday life practices, substance use, and performance expectations affect their health and well-being.

## 5. Conclusions

Our study suggests that when the “sober generation” grows older [4,10], it is likely that many of its members will start to drink, but not necessarily at such a high consumption level as earlier generations. Our analysis proposes multiple reasons for this. First, the changes in abstinence, moderate drinking, and heavy drinking trajectories among our young participants point out that all these trajectories in the maturing process over time become increasingly driven by relations that emphasise self-control, responsibility, and reflexivity. Secondly, the postponement of drinking to the legal age of 18 means that when young people start to drink as adults, they develop a relation to drinking that is more likely guided by elements of moderation than transgression [42]. Thirdly, as performance culture increasingly shapes young people’s abstinence and drinking trajectories, it also dampens their desire to develop heavy drinking practices. The changes we identify in the article are not only positive. In our data, there are also participants who relate their restrictive alcohol consumption to stress and pressure to perform well in studies, sports, work, and social relations. These increasing demands in a performance-oriented everyday life may have adverse effects on their overall health and social well-being.

## Figures and Tables

**Table 1 ijerph-19-03591-t001:** Drinking categories in the study (N = 28).

Alcohol Consumption	Wave 1/2017	Wave 2/2018	Wave 3/2019
Abstainers	15 (54%)	13 (46%)	9 (32%)
Moderate drinkers	7 (25%)	9 (32%)	8 (29%)
Heavy drinkers	6 (21%)	6 (21%)	11 (39%)
Total	28 (100%)	28 (100%)	28 (100%)

## Data Availability

Due to the nature of this research, participants of this study did not agree for their data to be shared publicly, so supporting data are not available.

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
