# Peer review of "Becoming Safe, Legal, Mature, Moderate, and Self-Reflexive: Trajectories of Drinking and Abstinence among Young People"

_ijerph, 2022, doi:10.3390/ijerph19063591_

Round 1

Reviewer 1 Report

I thank the authors for this longitudinal qualitative study of alcohol and abstinence among young people. In general, this study has been properly presented with a clear aim, methods and results achieved from a qualitative analysis.

Some aspects could be considered to improve this paper.

  1. The ‘drinking’ can be found across this manuscript. It seems that the ‘drinking’ is specifically used for ‘alcohol drinking’ in this study. To avoid possible confusion (drinking can be soft drinks), is it possible to check relevant parts and then correct them?
  2. Participants: it could be still unclear with the socioeconomic status of the young people who have been recruited in this study. Is it possible to provide with more details? In addition, what were the rationales to select these young people?
  3. Method: Alcohol was the key issue studied in this paper. It was not clear what types of alcohol can be touched for the participants. This question may come from the difference found between alcohol types: strong alcoholic drinks, less strong .., and beer (light type). It could not be denied that the type of alcoholic drinks would deliver various impacts.
  4. Section 2.4: a flow chart giving the functions and applications of NVivo in this analysis may be able to enhance the understanding of this part.

Author Response

Please see the attachment, the updated manuscript and the supplementary material 1, 2, 3.

Reviewer 2 Report

Samuelsson and colleagues present longitudinal qualitative interviews with a sample of young people to understand the trajectory of restricted youth drinking into adulthood. The authors adopt an assemblage thinking approach to understand the transformation of young people's drinking over a period of three years.

Overall, the paper is excellently and engagingly written. The methodology and analyses are appropriate, and excellently described. I only have a few minor suggestions:

  • On line 65, place a space before the word "to" in the following sentence "Alcohol consumption among 15-to 16-year-olds is well-studied".
  • Also place a space before the word "to" on line 67 in the following sentence: "The share of 16-to 29-year-olds in Sweden".
  • Lines 72-74, I would have loved another sentence to give an example of the different trajectories in youth drinking between Sweden and Australia.
  • The data presented in Table 1, did it stratify according to age? That is, were those who were abstainers in Wave 1 and converted to moderate or heavy drinkers more likely to be young? The authors won't have the numbers to do any formal analysis, of course, but I was curious if the abstainers skewed younger at wave 1.
  • Could the authors please add a comment on how the proportions in Table 1 relate to population data? That is, what are the population levels of abstainers, moderate and heavy drinkers for young people in Sweden, to provide a context for the behaviours reported herein.
  • The authors state that the study followed current ethical practices. Was the study formally reviewed by an ethics or institutional review committee?

Author Response

(The authors gave the same response as above.)

Reviewer 3 Report

Review of "Becoming safe, legal, mature, moderate and self-reflexive"

The paper explores the evolution of the relationship between young people and alcohol consumption into adulthood. To this end, the authors administer longitudinal qualitative interviews by studying the results obtained on 28 subjects over the 2017-2019 period.

Is the manuscript clear, relevant to the field, and presented in a well-structured way?

The paper is relevant and of interest to the scientific community as it explores the factors and constructs that regulate the relationship of young people with alcohol: The goal is to understand the widely demonstrated phenomenon of reduction in alcohol consumption compared to the previous generation. We also intend to investigate the habits that these subjects develop later in adulthood to define the evolution of their relationship with alcohol. Understanding the phenomenon and its evolution can pave the way towards increasing the well-being of society by identifying the factors and constructs that regulate alcohol consumption.

The paper is clear and the structure is adequate: The abstract introduces the paper by providing concisely the main information of the paragraphs; and the introduction, with a theoretical framework, presents the topic of the decline in alcohol consumption among young people and investigates the key concepts of "assemblage", "actant", "trajectory", and "chains of translations”. and methods” section defines precisely the instruments, the administration procedures and the sample adopted. Furthermore, it should be noted that the data have been encoded with the NVivo software.

The description of the qualitative analysis is lacking compared to the explanation of the procedures and the criteria adopted to produce the results, evidently the result of interpretation. Furthermore, it is requested to insert the protocol adopted containing the questions posed.

“The study is capable of both capturing the variation in drinking trajectories” (line 444) and justifies this statement.

The discussions propose a reflection on the results that have identified key concepts regulating alcohol consumption. Finally, the conclusions define the occurrence of a change in the behaviour of alcohol consumers in adulthood; however, the level of consumption does not reach that of previous generations.

Are the references cited current (mostly in the last 5 years)? Does it include an abnormal number of self-citations?

The number of references is adequate in consideration of the fact that the paper is not a review but an empirical study. It is also noted that most of the references come from recent years. The following reference is recommended for further information:

-Non-Drinkers’ Experiences of Drinking Occasions: A Population-Based Study of Social Consequences of Abstaining from Alcohol. Substance use & misuse57(1), 57-66.

  • Factors that predispose undergraduates to mental issues: A cumulative literature review for future research perspectives. Frontiers in Public Health, 10.

Is the manuscript scientifically valid, and is the experimental design appropriate to test the hypothesis?

A qualitative study is correctly adopted by means of a semi-structured interview, however the protocol and the description of the criteria for analysing the answers given are lacking.

Are the manuscript results reproducible based on the details provided in the methods section?

The "materials and methods" section clarifies the sample (but not its characteristics with precision) and the interview sessions (but not the criteria for analysing the answers given). It is therefore difficult to confirm the replicability of the study.

Are the figures/tables/images/schemes appropriate? Do they show the data correctly? Are they easy to interpret and understand? Are the data interpreted appropriately and consistently throughout the manuscript? Please include details of statistical analysis or data acquired from specific databases.

We use a single table that effectively highlights the categories of alcohol consumption associated with the subjects studied. The results derive from a qualitative and subjective interpretation. It is difficult to generalize the results as the sample size is small; therefore, we ask to outline the characteristics of the reference population in order to contextualize the results.

Are the conclusions consistent with the evidence and arguments presented?

The conclusions are consistent with the findings and arguments, and they provide a precise insight into the phenomenon of reducing alcohol consumption and its evolution in adulthood.

Please review ethical statements and data availability statements to ensure they are adequate.

The interview procedures followed current ethical guidelines that ensure voluntary participation, confidentiality and informed consent.

Author Response

(The authors gave the same response as above.)

Reviewer 4 Report

The paper reads well despite its a bit artificial language that introduces theoretical concepts such as  assemblage, actant. I am sure it could have been written in a simpler language without devoting its important theoretical virtues.

Despite this critical comment I like its content and have got just few suggestions for improvement and clarification.

I would suggest to extent a section on recruitment. As I understood, in recruitment, the authors considered not only demographic but also social background of participants but no information on their position in social structure is provided. It is neither clear why two thirds of participants were girls. Could you claim that gender does not matter in identification of trajectories? And that extending your sample of more boys could not extend your understanding of changes in drinking in a process of maturation?

I am missing a section on limitations of the study unless the authors are not aware of any limitations.

I would encourage use of a concept of high income countries at the expense of a vague concept of Western countries. Is Australia a Western country?

I agree that the clear decrease in drinking indicates globalisation of youth culture as much as previous increasing trend. Therefore, I suggest to claim that recent trends ‘confirm’ globalisation of youth culture rather than ‘indicate’ (40).

‘Structural changes in school and labour market’ need to be explained in more details (64).

A phrase ‘ … the decline of alcohol consumption among young adults in Sweden has stabilised …’ may suggest that the decline has continued or that the decline was replaced by stabilisation (71-72). Needs clarification.

How intoxication was defined for participants (133-134)?

In the second paragraph of discussion a question of illicit alcohol (bought from dealers) whose strength and composition are not known to young consumers could be added to justify a perception of drinking environment as unsafe (450-451).

The last sentence in the conclusion section needs clarification. Does it suggest that reduction in drinking may have adverse effects on their overall health and social wellbeing (523-525)?  

Author Response

(The authors gave the same response as above.)
